# Risk Stratification for Herpes Simplex Virus Pneumonia Using Elastic Net Penalized Cox Proportional Hazard Algorithm with Enhanced Explainability

**DOI:** 10.3390/jcm12134489

**Published:** 2023-07-05

**Authors:** Yu-Chiang Wang, Wan-Ying Lin, Yi-Ju Tseng, Yiwen Fu, Weijia Li, Yu-Chen Huang, Hsin-Yao Wang

**Affiliations:** 1Department of Medicine, Brigham and Women’s Hospital, Boston, MA 02115, USA; ywang134@bwh.harvard.edu; 2Department of Medicine, Harvard Medical School, Boston, MA 02115, USA; 3Syu Kang Sport Clinic, Taipei 112053, Taiwan; 4Department of Computer Science, National Yang Ming Chiao Tung University, Hsinchu 30010, Taiwan; 5Computational Health Informatics Program, Boston Children’s Hospital, Boston, MA 02115, USA; 6Department of Medicine, Kaiser Permanente Santa Clara Medical Center, Santa Clara, CA 95051, USA; 7Cardiovascular Institute, AdventHealth Orlando, Orlando, FL 32803, USA; 8Department of Thoracic Medicine, Chang Gung Memorial Hospital, Taipei 333, Taiwan; 9Department of Laboratory Medicine, Linkou Chang Gung Memorial Hospital, Taipei 333, Taiwan

**Keywords:** herpes simplex virus, pneumonia, risk stratification, elastic net penalized Cox proportional hazard algorithm, explainability

## Abstract

Herpes simplex virus (HSV) pneumonia is a serious and often fatal respiratory tract infection that occurs in immunocompromised individuals. The early detection of accurate risk stratification is essential in identifying patients who are at high risk of mortality and may benefit from more aggressive treatment. In this study, we developed and validated a risk stratification model for HSV bronchopneumonia using an elastic net penalized Cox proportional hazard algorithm. We analyzed data from a cohort of 104 critically ill patients with HSV bronchopneumonia identified in Chang Gung Memorial Hospital, Linkou, Taiwan: one of the largest tertiary medical centers in the world. A total of 109 predictors, both clinical and laboratory, were identified in this process to develop a risk stratification model that could accurately predict mortality in patients with HSV bronchopneumonia. This model was able to differentiate the risk of death and predict mortality in patients with HSV bronchopneumonia compared to the APACHE II score in the early stage of ICU admissions. Both hazard ratio coefficient and selection frequency were used as the metrics to enhance the explainability of the informative predictors. Our findings suggest that the elastic net penalized Cox proportional hazard algorithm is a promising tool for risk stratification in patients with HSV bronchopneumonia and could be useful in identifying those at high risk of mortality.

## 1. Introduction

HSV is a double-stranded DNA virus that is capable of infecting a variety of host tissues, including the central nervous system, respiratory tract systems, and the mucocutaneous system [1]. Transmission typically occurs through direct contact between oral-to-oral, oral-to-genital, genital-to-genital, or infected oral secretions [2]. Herpes infections are very common worldwide. In the United States, approximately 50–80% of adults have contracted the Herpes simplex virus (HSV) infection at least once during their lifetime. HSV infection remains largely asymptomatic, and only 20–25% of the infection develop clinical symptoms. The most common clinical manifestation of primary HSV infection is usually fever, malaise, headaches, sore throat, multiple herpetic gingivostomatitis, and localized lymphadenopathy [3,4]. HSV is a rather versatile organism that could infect immunocompetent or immunocompromised hosts. The infection of the upper respiratory tract is common and often self-limited, whereas the infection of the lower respiratory tract is more serious, causing bronchopneumonia mainly in immunocompromised or critically ill patients [4].

Due to difficulties in isolating HSV from the lower respiratory tract culture, the first characterization in the literature regarding HSV bronchopneumonia was guided solely by disease manifestations, with evidence of pulmonary parenchymal invasion and an unexplained clinical rapid decline [5]. In the present day, the universally accepted diagnostic testing of HSV bronchopneumonia is the polymerase chain reaction (PCR) test, which is performed using broncho-alveolar lavage (BAL) specimens from the lower respiratory tract [6]. Unfortunately, HSV bronchopneumonia is associated with a higher incidence of adult respiratory distress syndrome (ARDS) by up to 55% in comparison to 30% of all other organisms. Because of its extensive damage to the respiratory tract system, prolonged ventilator support use and increased mortality rate (approximately 45–60%) are often reported [5,7]. However, there is currently no evidence suggesting effective treatment for HSV bronchopneumonia. The benefit of the utilization of antiviral therapy, acyclovir remains unclear—some studies have shown benefits while others report no significant clinical improvement [5,7,8,9].

Given the fact that there is no reliable treatment for HSV bronchopneumonia, efforts have been made to identify its risk factors and correlations with mortality rates. Previous studies have identified the risk factors associated with increased mortality in patients diagnosed with HSV bronchopneumonia, including age, acute respiratory distress syndrome (ARDS), APACHE II score, and other comorbidities [10,11,12,13,14]. However, no previous studies have been able to provide the estimated contribution of each risk factor to the mortality rate of HSV bronchopneumonia. Additionally, most of these studies have used the univariable analysis method for its analysis, as it is unable to reflect the real clinical scenario, which usually involves multiple different factors in play. Therefore, the need to develop a more accurate, precise, and comprehensive method for HSV bronchopneumonia risk stratification is imperative and essential. Moreover, this method is vital in guiding clinical management due to its ability to identify patients at high risk, initiate proper or more aggressive treatment early on, and reduce mortality in these patients. Given the rapidly advancing machine learning algorithm, it has the power to analyze multivariable data in many medical fields, including obesity prediction [15], survival prediction for head-and-neck cancer patients [16], daily function prediction for brain stroke patients [17], and antibiotic resistance prediction [18], and is an excellent tool to use when investigating implicit patterns behind data. Thus, this study used machine learning and artificial intelligence to perform a comprehensive cross-group comparison and provide an accurate estimation of the mortality rate of HSV bronchopneumonia using the identified risk factors.

## 2. Materials and Methods

### 2.1. Study Design

We aimed to develop a predictive model that could provide risk stratification for patients with HSV bronchopneumonia. To be a clinically useful tool in a real-world setting, this model should avoid various biases in the training process. We used Elastic net penalized Cox (ENP Cox) proportional hazard regression with an internal feature penalty function to select informative features in the model training process. Additionally, we employed repeated nested cross-validation to evaluate the model’s robustness. To provide a clear explainability of the model, we estimated feature importance using both hazard ratio (HR) coefficients and selection frequency. In summary, we developed and validated an unbiased and understandable predictive model that could estimate the prognosis of HSV bronchopneumonia patients. The schematic illustration of this study design can be found in Figure 1.

### 2.2. Patients Eligibility

In this study, 119 adult patients (>18 years old) who had HSV detected from lower airway specimens were included (from January 2015 to January 2019). Demographic and clinical information, such as age, gender, smoking history, underlying diseases, laboratory test results, and history of steroid or immunosuppressive drug use, were recorded. Positive PCR or culture data were used to define HSV detection. The APACHE II score was used to assess the severity of patients. ARDS was defined based on the Berlin definition [19]. These HSV bronchopneumonia patients were divided into two groups: an alive group and a non-survival group based on the survival status in ICU admissions. The retrospective study was approved by the Institution Review Board (IRB) of CGMH (No. 202001331B0), and informed consent was waived on the basis of the retrospective study design. More detailed definitions for the information that was collected in this study can be found in a previously published study [20]. Fifteen patients were excluded due to missing data.

### 2.3. Model Development and Evaluation

The ENP Cox proportional hazard algorithm was used for training and validating a prognosis predictive model for HSV patients. The elastic-net mixing parameter (i.e., alpha) was set at 0.2. The prognosis-predictive model was trained and validated on the basis of the comprehensive features, including the clinical features and the lab tests (Table 1). The repeated nested cross-validation (NCV) method (five-fold) five times was used to robustly evaluate the models. The regulation parameter (i.e., lambda) was tuned in the inner layer of NCV. The performance of the models was evaluated in the outer layer of NCV. Given the settings of a five-fold CV repeated five times, there were 25 measurements in total. The survival index of the HSV patients was calculated in each iteration, and the median survival index was used as the cut-off to categorize the patients into high-risk and low-risk groups. The models were trained and tested using R software (version 3.5.2) with the glmnet package [16]. To evaluate the discriminative abilities of the ML model and APACHE II, the Harrell concordance index (C index) was used as the metric [21].

### 2.4. Features Importance

The importance of these features was estimated by two metrics. The first metric was the HR coefficients of the elastic net-Cox proportional hazard models. The second metric was the frequency of being selected in the models. Based on the design of model development and evaluation, 25 models were generated. The number of times they were selected for the features was used as the other metric to evaluate the feature’s importance.

### 2.5. Statistical Analysis

The data were presented as the mean ± standard deviation. Statistical significance between these two groups of continuous variables and categorical variables was determined using the *t*-test and Fisher’s exact test, respectively. The survival distributions of high-risk and low-risk groups were compared using a log-rank test. The threshold of the *p*-value was set as 0.5, where a *p*-value less than 0.05 was considered statistically significant. All data were analyzed using R, version 3.5.2 (R foundation for statistical computing).

## 3. Results

### 3.1. Baseline Clinical Characteristics of Subjects

Among all the 104 patients, 38 (36.5%) of them were survivors, and 66 (63.5%) were non-survivors. The patients included 76 (73%) males and 28 (27%) females. The mean age was 66.58 ± 13.16 in alive patients and 66.7 ± 15.5 in non-survival patients. The body mass index was 22.93 ± 4.4 in survivors and 23.08 ± 4.67 in non-survivors. There were 74 (71.2%) smokers or ex-smokers among the patients and 30 (28.8%) non-smokers. The mean APACHE II score was higher in non-survival patients (30.86 ± 5.7 vs. 25 ± 8.23, *p* < 0.001). More survivors had diabetes mellitus (DM) (15 (39.5%) vs. 3 (4.5%), *p* < 0.001), but fewer alive patients had an acute kidney injury (AKI) (13 (34.2%) vs. 52 (78.8%), *p* < 0.001). In terms of ventilation settings, the mean of the fraction of inspired oxygen (FiO_2_) was higher in non-survivors (54.55 ± 20.01% vs. 44.47 ± 13.35%, *p* = 0.007), and the mean of the A-a gradient was also higher in non-survivors (230.33 ± 133.89 vs. 178.96 ± 106.36, *p* = 0.046). However, tidal volume, positive end-expiratory pressure (PEEP) and positive inspiratory pressure (PIP) showed no significant differences between the two populations. Most of the non-survivors developed ARDS (8 (21.1%) vs. 46 (69.7%)) during the same admission (Table 1).

### 3.2. Risk Stratification Based on Different Indicators

A comparison between the survival probability of high-risk patients and low-risk patients are presented in Figure 2A,B by using two different discrimination methods. In Figure 2A, the survival probability (in percentage) of high-risk (red) and low-risk (blue) patients were analyzed using the ENP Cox model during their days in the ICU, with correspondence of a 95% confidence interval painted in the shaded area. The survival probability decreased over time, indicating that the two groups’ mortality risk increased over time. However, the high-risk group had a significantly lower survival probability (*p* = 0.003) than the low-risk group, which started at the very beginning of their ICU stay.

More significantly, while comparing the 95% confidence intervals (shaded region) of the survival curves between Figure 2A,B, it demonstrated that there was less overlapping in the shaded area when using the ENP Cox model (Figure 2A) than the APACHE II score (Figure 2B), with a more notable separation between the high-risk and low-risk groups in the early phase of their ICU stay (<50 days of ICU stay) when using the ENP Cox model. No significance was found between high-risk and low-risk patients, and constant overlapping throughout the shaded regions for both curves were noted when the APACHE II score was used for analysis. In addition, the median survival time for the high-risk group was 24 days, and for the low-risk group was 46 days in Figure 2A. In Figure 2B, the median survival time for both high- and low-risk groups were noted to be similar, both around 37 days. Impressively, the ENP Cox model demonstrated a higher classification performance than APACHE II when identifying high-risk patients (mean (SD) C index, 0.624 (0.093) vs. 0.603 (0.093); *p* < 0.001).

### 3.3. Risk Factor Identification

The elastic net Cox regression model was able to estimate the importance of a specific feature contributing to the survival curve using HR and selection frequency, as shown in Figure 3 and Table 2. The selection frequencies were presented when a feature was selected more than 50% of the time. FiO2 was the most informative feature and was noted with a 100% selection frequency before a subsequently atypical lymphocyte percentage (96%) and APACHE II (92%) in Figure 3. HRs were also noted in Figure 3 and Table 2. Nonetheless, despite a significantly high selection frequency for FiO2, the HR of FiO2 was 1.01 (0.01). Other notable data findings included the HR of steroids, which was 0.81 (<1.0), and the HR of combined bacteria, which was 1.14 (>1.0).

## 4. Discussion

The significance of critically ill patients with HSV bronchopneumonia who survived advanced treatments has recently gained the attention of ICU physicians. The population of patients with this condition is growing and is relatively new [22], making their precise stratification an unmet need. In our study, we aimed to develop a risk stratification model that could identify HSV bronchopneumonia patients with a high risk of mortality. To create a model that is unbiased and clinically useful, we needed to avoid overfitting and provide explainability. We used the ENP Cox proportional hazards regression with an internal feature penalty function to select informative features in the training process (Figure 1). Additionally, we employed repeated nested cross-validation to evaluate the models in an unbiased manner (Figure 1). Our results showed that this model could effectively discriminate between high-risk and low-risk patients. According to the repeated nested cross-validation, the selection frequency in multiple iterations of the cross-validation also provided valuable information regarding feature importance, in addition to the HR coefficient. This model could predict the risk of HSV bronchopneumonia in patients and provide a key for better management and treatment clinically.

A total of 109 clinical features were collected for the patients in our study. In the univariable analysis, several features showed significant differences between the alive and non-survival groups (Table 1). Not surprisingly, the APACHE II score, a widely used ICU score, was significantly higher in the non-survival group than in the alive group. Similar trends were found for the percentages of ARDS, FiO2, and AKI. These findings indicated that poor lung and renal functions were associated with a higher mortality rate of HSV bronchopneumonia. The percentage of DM was also significantly different between the alive and non-survival groups. However, DM was associated with survival rather than mortality. This finding was somewhat counterintuitive, as DM is typically considered a risk factor for many diseases [23]. In an observational prospective multicenter study, DM was associated with a lower rate of developing ARDS in critically ill patients [24]. The protective effect of DM in preventing ARDS development remained even when factors such as blood sugar levels or anti-glycemic agents were adjusted [24]. DM was also found to be associated with a lower mortality rate in sepsis patients [25]. Although its underlying mechanism is not well understood, the protective effect of DM may be due to DM-associated hyperglycemia or an altered metabolism that is not seen in non-DM patients.

As shown above, the ENP Cox model can effectively distinguish high-risk patients from low-risk patients (Figure 2A). The mortality risk over time for the two groups (i.e., high-risk and low-risk) showed significant differences. The same trend was also found when we used the APACHE II score alone as the predictor to discriminate between the high-risk and low-risk groups (Figure 2B). Both the ENP Cox model with a full set of features and the APACHE II score showed an ability to discriminate between high-risk and low-risk patients. However, the survival curves in the two discrimination methods demonstrated different patterns. Briefly, there was less overlap in the 95% confidence intervals of the survival curves when using the ENP Cox model (Figure 2A) for classification than when using the APACHE II score alone (Figure 2B). The separation of the survival curves was more obvious in the early phase of the disease course (i.e., <50 days of ICU stay). When the ENP Cox model was used to classify the patients, the median survival time for the high-risk and low-risk groups was 24 days and 46 days, respectively. The median survival time of the low-risk group was around two times that of the high-risk group. By contrast, the median survival time for both the high-risk and low-risk groups was around 37 days (Figure 2B). These results indicate that the ENP Cox model is a more adequate clinical tool than using the APACHE II score alone in detecting high-risk HSV bronchopneumonia patients, especially in the early stages of the disease course.

The explainability of a predictive model is important due to its predictive performance. We adopted both conventional methods (i.e., univariable analysis) and ENP Cox algorithm embedded methods (i.e., penalized methods) when estimating feature importance. The discordance between these two methods was obvious, where some risk factors, such as DM, ARDS, and AKI in univariable analysis (in Table 1), were not ranked as top informative features in the ENP Cox model (Figure 3). By contrast, informative features of the ENP Cox model, such as the atypical lymphocyte percentage, lymphocyte counts, and patient height, were not regarded as predictors for discriminating between the alive and non-survival groups. To better fit clinical use in a real-world setting, the feature importance derived from the ENP Cox algorithm and repeated nested cross-validation would likely be more adequate in prediction and future use. The first reason for this is that weak features in the ENP Cox algorithm were penalized and removed in the competition of features, thus eliminating a confounding bias in the model learning process. Moreover, the overfitting bias caused by information leakage could be avoided by using nested cross-validation. Based on the central limit theorem, a repeated process of sampling and cross-validation could provide additional benefits when estimating true statistics. These advantages are particularly important for estimating the true characteristics of the real world using small datasets.

In addition to the traditional HR coefficient, we also used selection frequency as another indicator of feature importance. All the HR coefficients of the informative features were around one (see Table 2), indicating that there was no predominantly strong predictor in the ENP Cox model. Additionally, the HR coefficients between the informative features were similar, making it difficult to determine their relative importance. By contrast, the selection frequency of features in the ENP Cox model could serve as an alternative metric to estimate feature importance. For example, when FIO2 was selected in all model training iterations (100%) as an informative feature (Figure 3), the HR coefficient of FIO2 was not significantly higher compared to the other informative features (Table 2). Therefore, combining both metrics could be a more comprehensive way to estimate the impact of features.

The top five informative features in the ENP Cox model were FIO2, atypical lymphocytes, APACHEII, height, and lymphocytes. All of these features are risk factors for mortality in HSV pneumonia. FIO2 is the level of oxygenation that is used to maintain adequate oxygen saturation. When higher FIO2 is needed, the inhaled oxygen is difficult to transfer from the alveoli into the blood, thus indicating more severe lung destruction. Higher mortality associated with higher FIO2 is accordingly reasonable. Reactive lymphocytosis, as well as atypical lymphocytosis, can be potential prognostic markers for HSV pneumonia. Previous studies have demonstrated the diagnostic and prognostic value of lymphocytosis in various pulmonary conditions, such as hypersensitivity pneumonitis and lymphocytic interstitial pneumonia [26]. However, the significance of blood lymphocytosis in the prognosis of HSV pneumonia remains relatively unexplored. Body height was also found to be a risk factor for HSV pneumonia in the ENP Cox model. Body height has been associated with poor prognosis in various lung diseases, including COVID pneumonia [27], chronic obstructive pulmonary disease [28], and non-small cell lung cancer [29]. The association between body height and pneumonia prognosis could be understood in the context of the overall health status of individuals. The distribution of the pectoralis muscle per height square reduces with a taller body height and can impact the severity and recovery from pneumonia [27].

A risk stratification model for HSV bronchopneumonia that can predict the duration of hospitalization, complications during hospitalization, recovery duration, and final outcomes would be clinically useful. In comparison to the well-established APACHE II Score model, the results show that our model for HSV bronchopneumonia could better predict the duration of hospitalization, complications during hospitalization, recovery duration, and final outcomes. This risk stratification model is revolutionary and significantly more useful clinically. Not only would this model be better able to guide clinical management and treatment plans, but clinicians would also be able to utilize the results from this model and direct the goals of care discussion amongst patients and their family members. However, based on the dataset we used in this study, survival time is the only outcome measurement we have. Thus, the ML model can only predict survival time according to the dataset. The extrapolation of the ML model to other outcomes is inadequate in this model. Therefore, predictions on other clinically meaningful outcomes are worthy of further investigation in specified ML models.

## 5. Conclusions

We developed a risk stratification model to identify patients with HSV bronchopneumonia who are at high risk of mortality. This model adopted the ENP Cox algorithm with an internal feature penalty function to select informative features in the training process and repeated nested cross-validation to evaluate the models in an unbiased manner. The results showed that this model could effectively discriminate between high-risk and low-risk HSV bronchopneumonia patients. This model could be useful in predicting patient outcomes and improving clinical management and the treatment of patients with HSV bronchopneumonia.

## Figures and Tables

**Figure 1 jcm-12-04489-f001:**
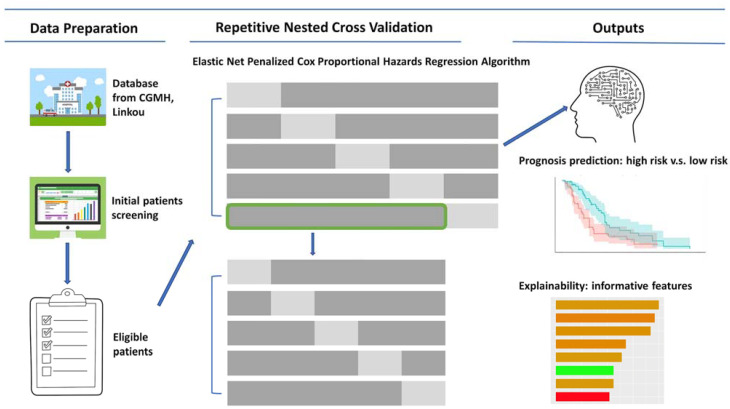
Scheme of the study. We adopted three steps, including data preparation, repetitive nested cross-validation, and the evaluation of model outputs. In data preparation, critically ill patients (ICU admission, *n* = 119) with HSV bronchopneumonia were enrolled in initial patient screening. Cases with missing data were excluded, and eligible cases (*n* = 104) were used for the following analyses. In model training and validation, ENP Cox proportional hazards regression algorithm-based models were trained and validated by the repetitive nested cross-validation method (Dark Gray color represents samples selected by the machine learning process, while Light Gray color represents samples that were not selected). A built-in property of the ENP Cox proportional hazards regression algorithm could be used for feature selection. Moreover, unbiased estimation of model performance was possible through the repetitive nested cross-validation method. In the outputs, the model performance on risk stratification for HSV bronchopneumonia patients was revealed. Additionally, selection frequency and the hazard ratio coefficient of the features were both used to enhance the explainability of the informative features in the predictive model.

**Figure 2 jcm-12-04489-f002:**
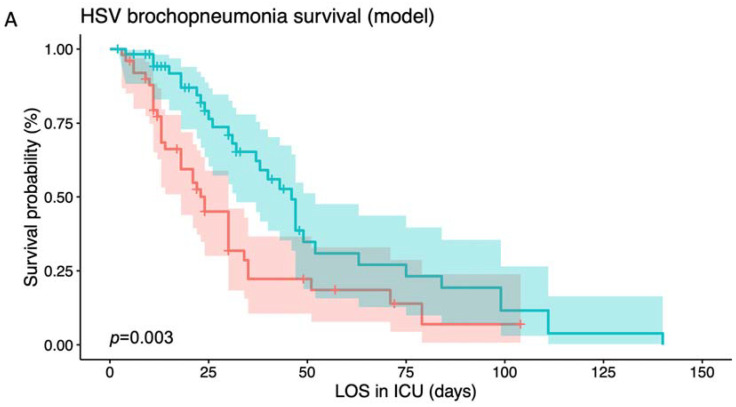
Risk stratification based on different indicators. (**A**). Survival curves of high-risk (red) and low-risk (green) patients using an elastic net Cox regression model. The high-risk group could be discriminated against from the low-risk group (*p* = 0.003). Of note, the discrimination between the high-risk and low-risk groups is obvious in the relatively early phase (<50th day), where the two survival curves do not overlap with each other. (**B**). Survival curves of high-risk (red) and low-risk (green) patients using an APACHE II score. The survival curves of the high-risk and low-risk groups are statistically distinguishable (*p* = 0.044). However, 95% confidence intervals of the survival curves overlap with each other throughout the whole phase. LOS in ICU: length of stay in intensive care unit.

**Figure 3 jcm-12-04489-f003:**
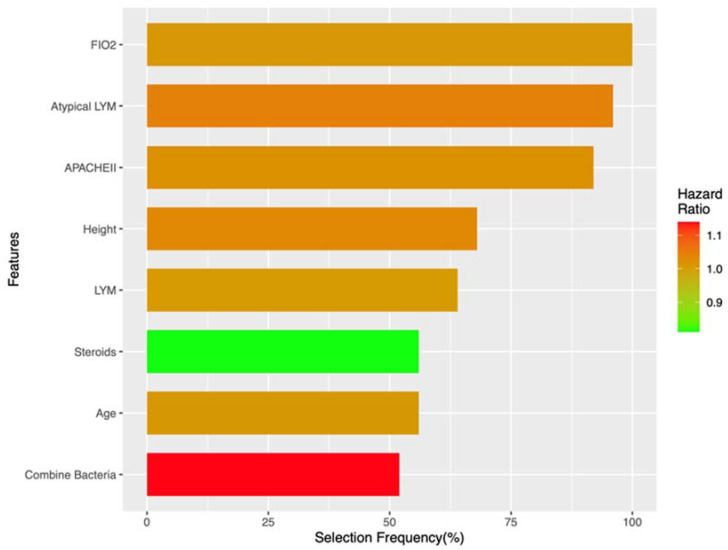
Feature importance of elastic net Cox regression model.

**Table 1 jcm-12-04489-t001:** Patient characteristics of alive and non-survival groups.

Feature Name	Level	Alive (*n* = 38)	Non-Survival (*n* = 66)	*p*
Gender (%)	F	11 (28.9)	17 (25.8)	0.902
	M	27 (71.1)	49 (74.2)	
Age (mean (SD))		66.58 (13.16)	66.70 (15.50)	0.968
Height (mean (SD))		162.26 (5.90)	162.45 (6.30)	0.882
BW (mean (SD))		60.25 (11.42)	60.90 (12.43)	0.792
BMI (mean (SD))		22.93 (4.40)	23.08 (4.67)	0.876
Smoking (%)	NO	12 (31.6)	18 (27.3)	0.809
	YES	26 (68.4)	48 (72.7)	
Oral Ulcer (%)	NO	18 (47.4)	27 (40.9)	0.664
	YES	20 (52.6)	39 (59.1)	
APACHE II (mean (SD))		25.00 (8.23)	30.86 (5.70)	<0.001
CRP (mean (SD))		116.74 (82.02)	122.11 (108.60)	0.792
WBC (mean (SD))		10,828.95 (5247.49)	12,342.42 (8524.54)	0.324
Lymphocyte percentage (mean (SD))		8.21 (5.43)	8.78 (14.10)	0.812
Lymphocyte count (mean (SD))		817.43 (614.34)	618.85 (587.21)	0.106
Atypical lymphocyte percentage (mean (SD))		0.31 (0.66)	1.21 (4.64)	0.24
HSV alone (%)	NO	28 (73.7)	57 (86.4)	0.178
	YES	10 (26.3)	9 (13.6)	
Bacteria combined (%)	NO	16 (42.1)	24 (36.4)	0.711
	YES	22 (57.9)	42 (63.6)	
Fungus combined (%)	NO	32 (84.2)	48 (72.7)	0.273
	YES	6 (15.8)	18 (27.3)	
Aspergillosis (%)	NO	37 (97.4)	58 (87.9)	0.195
	YES	1 (2.6)	8 (12.1)	
Combine Virus (%)	NO	30 (78.9)	42 (63.6)	0.159
	YES	8 (21.1)	24 (36.4)	
PJP (%)	NO	35 (92.1)	53 (80.3)	0.185
	YES	3 (7.9)	13 (19.7)	
Mycobacterium spp. (%)	NO	38 (100.0)	64 (97.0)	0.732
	YES	0 (0.0)	2 (3.0)	
Organ failure (%)	NO	32 (84.2)	43 (65.2)	0.063
	YES	6 (15.8)	23 (34.8)	
Diabetes mellitus (%)	NO	23 (60.5)	63 (95.5)	<0.001
	YES	15 (39.5)	3 (4.5)	
Immunocompromise (%)	NO	22 (57.9)	27 (40.9)	0.142
	YES	16 (42.1)	39 (59.1)	
Sepsis (%)	NO	23 (60.5)	53 (80.3)	0.05
	YES	15 (39.5)	13 (19.7)	
Cardiovascular Crisis (%)	NO	35 (92.1)	65 (98.5)	0.271
	YES	3 (7.9)	1 (1.5)	
CXR.GGO (%)	NO	33 (86.8)	59 (89.4)	0.941
	YES	5 (13.2)	7 (10.6)	
CXR.Interstitial (%)	NO	28 (73.7)	45 (68.2)	0.713
	YES	10 (26.3)	21 (31.8)	
CXR.Consolidation (%)	NO	15 (39.5)	31 (47.0)	0.592
	YES	23 (60.5)	35 (53.0)	
CT.GGO (%)	NO	37 (97.4)	63 (95.5)	1
	YES	1 (2.6)	3 (4.5)	
CT.Interstitial (%)	NO	33 (86.8)	57 (86.4)	1
	YES	5 (13.2)	9 (13.6)	
CT.Consolidation (%)	NO	26 (68.4)	50 (75.8)	0.56
	YES	12 (31.6)	16 (24.2)	
Bronchoscopy (%)	NO	12 (31.6)	20 (30.3)	1
	YES	26 (68.4)	46 (69.7)	
ARDS (%)	NO	30 (78.9)	20 (30.3)	<0.001
	YES	8 (21.1)	46 (69.7)	
AKI (%)	NO	25 (65.8)	14 (21.2)	<0.001
	YES	13 (34.2)	52 (78.8)	
Steroids (%)	NO	19 (50.0)	26 (39.4)	0.398
	YES	19 (50.0)	40 (60.6)	
Treatment (%)	NO	30 (78.9)	49 (74.2)	0.762
	YES	8 (21.1)	17 (25.8)	
Treat enough (%)	NO	32 (84.2)	50 (75.8)	0.443
	YES	6 (15.8)	16 (24.2)	
PEEP (mean (SD))		9.32 (1.76)	9.36 (1.76)	0.894
delta P (mean (SD))		16.29 (5.02)	18.61 (6.08)	0.049
PIP (mean (SD))		25.87 (5.79)	27.82 (6.46)	0.127
TV (mean (SD))		486.13 (88.36)	468.61 (104.29)	0.386
FIO2 (mean (SD))		44.47 (13.35)	54.55 (20.01)	0.007
A-a gradient (mean (SD))		178.96 (106.36)	230.33 (133.89)	0.046

**Table 2 jcm-12-04489-t002:** Coefficients of the important features in the elastic net Cox regression model.

Feature Type	Feature Name	Selection Frequency (%)	HR (SD)
Risky factor	FIO2	100	1.01 (0.01)
Risky factor	Atypical lymphocyte percentage	96	1.04 (0.03)
Risky factor	APACHEII	92	1.02 (0.02)
Risky factor	Height	68	1.03 (0.05)
Risky factor	Lymphocyte percentage	64	1.01 (0.01)
Risky factor	Age	56	1.01 (0.01)
Protective factor	Steroids	56	0.81 (0.14)
Risky factor	Bacteria combined	52	1.14 (0.21)

HR: hazard ratio coefficient. SD: standard deviation.

## Data Availability

The data presented in this study are available on request from the corresponding author. The data are not publicly available due to privacy.

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
