# Peer review of "Risk Stratification for Herpes Simplex Virus Pneumonia Using Elastic Net Penalized Cox Proportional Hazard Algorithm with Enhanced Explainability"

_jcm, 2023, doi:10.3390/jcm12134489_

Round 1
Reviewer 1 Report
Because Herpes Simplex Virus (HSV) pneumonia is a serious and fatal respiratory infection that occurs in immunocompromised individuals, truly accurate risk stratification would be invaluable for these patients. Making this classification early, on the other hand, will completely change the clinical course by identifying patients at high risk of death and who may benefit from more aggressive treatment. It s a well designed algorithm and this really would be very helpfull to the clinicians. But the introduction and the need such an algorithm could be more identified.
Author Response
Response:
Thank you for the recognition of the importance of accurate risk stratification for HSV pneumonia. To justify the use of ML algorithms, we have added some contents explaining the necessity. Please refer to the paragraph as follows and page 2 of the revised manuscript.
“Therefore, the need for development of a more accurate, precise, and comprehensive method for HSV bronchopneumonia risk stratification is imperative and essential. Moreover, this method would be vital in guiding clinical management due to the ability in identifying patients of high risk, initiating proper or more aggressive treatment early on, and reducing mortality in those patients. Given the rapidly advancing machine learning algorithm, it has the power of analyzing multivariable data in many medical fields, including obesity prediction, survival prediction for head-and-neck cancer patients, daily function prediction for brain stroke patients, and antibiotic resistance prediction, and is an excellent tool to use to investigate implicit patterns behind data. Thus, this study used machine learning and artificial intelligence to perform a comprehensive cross-group comparison and provide an accurate estimation the mortality rate of HSV bronchopneumonia using the identified risk factors.”

Reviewer 2 Report
The manuscript by Wang, et al entitled “Risk stratification for herpes simplex virus pneumonia using elastic net penalized Cox proportional hazard algorithm with enhanced explainability” explores the design of modeling to assist in the prediction of mortality in those with pneumonia due to herpes simplex virus (HSV). The study makes a comparison to mortality estimates derived from the acute physiologic and chronic health evaluation II (APACHE II), a well validated and accepted determination of mortality in critically ill patients. The study is well designed with the sample size seemingly appropriate in the context of a rarer form of pneumonia and inducer of the acute respiratory distress syndrome (ARDS). Some of the results, such as correlation of mortality with the number of organ failures, development of ARDS, and acute kidney injury are not surprising nor novel. The major concern of this reviewer is how the developed model is superior to APACHE II in assessing mortality in this patient population and how this model will be useful in the care and management of patients with severe acute respiratory failure due to HSV. This is only partially discussed in the manuscript. A model predicting duration of hospitalization, complications during hospitalization, recovery duration, and final outcomes (home, long-term acute care/rehabilitation, or nursing home) would be clinically useful in discussion with patients and their family members.
Additional concerns from this reviewer:
Minor
1. Better terms for “Survived” and “Mortality” would be “Alive” and “Non-survived”
2. On Page 6, line 145 there is a typo regarding BMI for survivors; listed as 60.25. Additional typos are on Page 1, line 23 “ill” for “illed” and on Page 7, line 170 in the legend for Figure 2; “early” for “ealy”.
Author Response
Response:
1) We totally agree with your professional comments that the identified important features are not surprising nor novel. In contrast, the high agreement between the important features of the ML model and the previous publications indicates that the prediction or classification are derived from reasonable factors. Moreover, using both selection frequency and HR coefficients has reinforced the explainability of the ML model. Knowing the actual predictors used by the ML model to generate the prediction is a key to a useful ML model for clinical use because the kind of ML model is not an unexplainable black box even with high predictive performance.
2) Regarding the superiority of the ML model over APACHE II in assessing mortality in this patient population, we added Harrell concordance index (C index) as the metric to access the discriminative abilities of the ML model and APACHE II. The related materials&methods and results of using C index have been added to the revised manuscript. Please refer to the follows:
M&M: “To evaluate the discriminative abilities of the ML model and APACHE II, Harrell concordance index (C index) was used as the metric.” (Page 4)
Results: “The ENP Cox model demonstrated higher classification performance than APACHE II in identifying high risk patients (mean [SD] C index, 0.624 [0.093] vs 0.603 [0.093]; P < 0.001).” (Page 8)
3) We agree with you that a model predicting duration of hospitalization, complications during hospitalization, recovery duration, and final outcomes would be clinically useful. However, based on the dataset we used in the study, survival time is the only outcome measurement we have. Thus, the ML model can only predict survival time according to the dataset. Extrapolation of the ML model to other outcomes is inadequate and would generate inaccurate results. Indeed, predictions on these outcomes including duration of hospitalization, complications during hospitalization, recovery duration, and final outcomes would be clinically useful and worthy of developing further specified ML models.
We put the discussion into the revised manuscript. Please refer to Page 10.
“A risk stratification model for HSV bronchopneumonia that can predict duration of hospitalization, complications during hospitalization, recovery duration, and final out-comes would be clinically useful. In comparison to the well-established APACHE II Score model, the results show that our model for HSV bronchopneumonia can better predict duration of hospitalization, complications during hospitalization, recovery duration, and final outcomes. This risk stratification model is revolutionary and significantly more useful clinically. Not only will this model be able to better guide clinical management and treatment plan, but clinicians would also be able to utilize the results from this model and direct goals of care discussion amongst patients and their family members. However, based on the dataset we used in the study, survival time is the only outcome measurement we have. Thus, the ML model can only predict survival time according to the dataset. Extrapolation of the ML model to other outcomes is inadequate in this model. Therefore, predictions on other clinical meaningful outcomes is worthy of further investigation in specified ML models.”
4) Regarding the additional concerns raised, we have modified the wordings and corrected typos thorough the manuscript according to your suggestion.

Reviewer 3 Report
The authors report on the utilization of the Elastic Net Penalized (ENP) Cox Proportional Hazard Algorithm to predict the risk factors associated with herpes simplex bronchopneumonia in patients. The authors provide a strong rationale for the study. While the cohorts are relatively small, the comparison between the ENP Cox algorithm and the APACHE II test provides a strong argument for the utilization of the approach for predictive outcomes of infection. The data are clearly presented, and the conclusions well supported by the findings.
Minor Points:
The authors compare the body mass indexes between the survivor and mortality groups on line 145 of the manuscript. However, the number reported for the survivors (60,25) is actually the mean body weight of the group. The correct number, from Table 1, should be 22.93.
A better explanation of the parameters chosen for the analysis would be helpful in better understanding the significance of the study.
The Quality of English is very high. A few minor points (such as the use of "illed" patients in the Abstract.
Author Response
1) We have corrected the typo. Thank you for the meticulous review!
2) We have a paragraph addressing the top informative parameters that are included in the ENP Cox model. Please refer to Page 10.
“The top 5 informative features in the ENP Cox model are FIO2, atypical lymphocyte, APACHEII, height, and lymphocyte. All of the features are risky factors to mortality of HSV pneumonia. FIO2 is the level of oxygenation that is used to maintain adequate oxygen saturation. When higher FIO2 is needed, the inhaled oxygen is difficult to be transferred from alveoli into blood and indicating a more severe lung destruction. A higher mortality associated to higher FIO2 is reasonable accordingly. Reactive lymphocytosis as well as atypical lymphocytosis can be the potential prognostic markers for HSV pneumonia. Previous studies have demonstrated the diagnostic and prognostic value of lymphocytosis in various pulmonary conditions, such as hypersensitivity pneumonitis and lymphocytic interstitial pneumonia. However, the significance of blood lymphocytosis in the prognosis of HSV pneumonia remains relatively unexplored. Body height is also found to be a risk factor of HSV pneumonia in the ENP Cox model. Body height has been associated with poor prognosis in various lung diseases, including COVID pneumonia, chronic obstructive pulmonary disease, and non-small cell lung cancer. The association between body height and pneumonia prognosis can be understood in the context of the overall health status of individuals. Distribution of pectoralis muscle per height square reduces with taller body height, and can impact the severity and recovery from pneumonia.”

Round 2
Reviewer 1 Report
I want to thank you for the detailed explanation.